# What Makes Partial-Label Learning Algorithms Effective?

**Jiaqi Lv**[1,5], **Yangfan Liu**[1], **Shiyu Xia**[1], **Ning Xu**[1,5], **Miao Xu**[2],
**Gang Niu**[3,1], **Min-Ling Zhang**[1,5], **Masashi Sugiyama**[3,4], **Xin Geng**[1,5*]
[1]Southeast University    [2]The University of Queensland
[3]RIKEN Center for Advanced Intelligence Project    [4]The University of Tokyo
[5]Key Laboratory of New Generation Artificial Intelligence Technology and
Its Interdisciplinary Applications (Southeast University), Ministry of Education, China
{is.jiaqi.lv, gang.niu.ml}@gmail.com,
miao.xu@uq.edu.au, sugi@k.u-tokyo.ac.jp,
{liuyangfan, shiyu_xia, xning, zhangml, xgeng}@seu.edu.cn

## Abstract

A *partial label* (PL) specifies a set of candidate labels for an instance and *partial-label learning* (PLL) trains multi-class classifiers with PLs. Recently, many methods that incorporate techniques from other domains have shown strong potential. The expectation that stronger techniques would enhance performance has resulted in prominent PLL methods becoming not only highly complicated but also quite different from one another, making it challenging to choose the best direction for future algorithm design. While it is exciting to see higher performance, this leaves open a fundamental question: *what makes a PLL method effective?* We present a comprehensive empirical analysis of this question and summarize the success of PLL so far into some *minimal algorithm design principles*. Our findings reveal that high accuracy on benchmark-simulated datasets with PLs can misleadingly amplify the perceived effectiveness of some general techniques, which may improve representation learning but have limited impact on addressing the inherent challenges of PLs. We further identify the common behavior among successful PLL methods as a progressive transition from uniform to one-hot pseudo-labels, highlighting the critical role of *mini-batch PL purification* in achieving top performance. Based on our findings, we introduce a *minimal working algorithm* that is surprisingly simple yet effective, and propose an improved strategy to implement the design principles, suggesting a promising direction for improvements in PLL.

## 1 Introduction

*Partial-label learning* (PLL) [14, 12, 42] has been an established discipline in the weakly supervised learning field [50, 51] for decades. It aims to train multi-class classifiers from instances with *partial-labels* (PLs)—a PL for an instance is a set of candidate labels, where a *fixed but unknown* candidate is the true label.

As benchmarking on simulated PLL versions of vision datasets becomes standard practice for evaluating PLL methods, new PLL approaches are emerging that integrate diverse advanced techniques to enhance the performance. While many methods show great promise and some headway has been made in understanding the methodology against ambiguous label assignments [24], we find that state-of-the-art (SOTA) approaches *look quite complicated and differ significantly from each other* (Table 1), making it challenging to choose the best direction for better algorithm design.

---

*Corresponding author.

38th Conference on Neural Information Processing Systems (NeurIPS 2024).

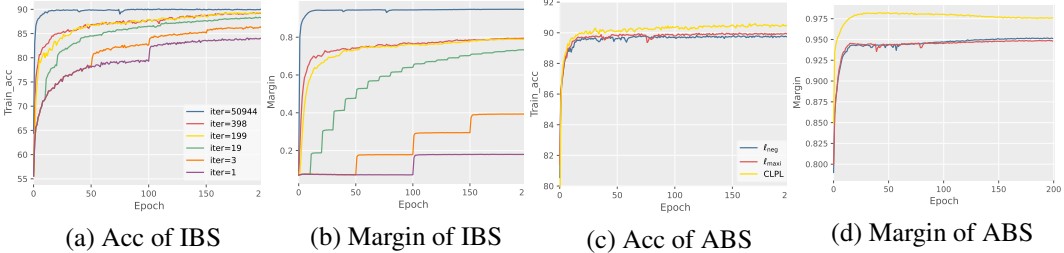

| (a) Acc of IBS | (b) Margin of IBS | (c) Acc of ABS | (d) Margin of ABS |

Figure 1: Training accuracy and confidence margin of predicted pseudo-labels for traditional IBS and ABS methods on FMNIST with PLs: (a-b) show the granularity of EM execution in classical IBS methods is refined from a single step to an entire epoch, facilitating a smoother transition from uniform to one-hot pseudo-labels; (c-d) demonstrate that when ABS methods are optimized using SGD, the optimization targets for candidate labels can gradually become distinct.

In this paper, we aim to summarize the success of PLL so far, and understand the indispensable elements in a well-performing PLL method, which can be condensed into *minimal algorithm design principles*. For this purpose, we think of the taxonomy of PLL approaches first, comb the evolution and trends in this field, thereby uncovering key factors that drive their effectiveness and motivating future research.

The widely accepted PLL taxonomy [47, 24] divides methods based on the treatment of PLs into *identification-based strategy* (IBS) and *average-based strategy* (ABS):
• IBS disambiguates each PL to select the most likely true label for training;
• ABS treats all candidate labels equally for training.
However, the boundary between these two categories is vague, resulting in a lack of consensus on the category of many recent approaches which *purifies each PL on the fly during model training* [25, 37, 39]. They initially look like ABS, since uniform targets are always used to prepare for true-label selection, and as training progresses, the optimization targets for candidate labels become distinct gradually, such that they exhibit IBS-like characteristics. The tricky fact is that such approaches are classified as ABS or IBS hinging on which definition of category researchers are willing to relax (e.g., [45] versus [35])! In light of the evolving approaches within PLL, is there a need to establish a third category within the taxonomy to capture the "hybrid strategy", as attempted by recent work like [43, 5]?

We suggest that the answer is in the negative, since our analysis confirms the fluidity of method categorization within PLL, emphasizing that IBS, ABS, and so-called hybrid strategy often overlap due to the dynamic nature of their implementation:
• Training manner of IBS. Typical IBS approaches [48, 13] perform one-step EM to identify the true label in each PL (E step) and then train a supervised classifier (M step). To mitigate overfitting to wrongly identify labels, multi-step EM IBS [11, 10] are proposed. As iterations are executed more frequently, the identifocation of true labels becomes increasingly smooth (Figure 1).
• Implicit differentiation in ABS. Typical ABS approaches [19, 6] design a loss function that does not differentiate candidate labels, e.g., enforcing the softmax outputs of them to sum to 1 [19]. However, stochastic optimization algorithms can *implicitly* lead to distinct outputs for each candidate label and show a progressive purification characteristic (Figure 1 and Lemma 3.1).
• Execution matters. Whether a method truly exhibits progressive characteristics and the extent of progression also depend on the optimizers used and hyperparameters like the learning rate and batch size (Figure 4).
Therefore, predefining a method's category and then asserting its utility can limit our understanding of its true nature and effectiveness. Further, our empirical incestigations reveal a key insight: all successful PLL algorithms exhibit a common behavior characterized by a *progressive transition from uniform to one-hot pseudo-labels*, facilitated by the combination of PL purification and model updates in a *mini-batch-wise manner*. Building on this core strategy, modern methods also integrate various techniques from other domains, exemplified by Match from semi-supervised learning [3, 29], aiming to further enhance model performance. While intuitively, better performance could be achieved if stronger techniques are employed, our findings indicate that these enhancements often yield marginal gains when compared to the primary benefits derived from mini-batch PL purification. Notably, even

Table 1: Comparison of techniques used in eight prominent PLL methods. $\checkmark/\times$ indicates whether a technique is used, and an underline denotes the key components of the respective methods.

| PLL methods | Mini-b. purif. | Mixup | Data augment. | Exp.Mov Average | Match DA | Match DM | Main assumption |
|---|---|---|---|---|---|---|---|
| PRODEN [25] | $\underline{\checkmark}$ | $\times$ | $\times$ | $\times$ | $\times$ | $\underline{\checkmark}$ | DNNs learn pattern first |
| CC [12] | $\checkmark$ | $\times$ | $\times$ | $\times$ | $\times$ | $\times$ | PLs are generated uniformly |
| LWC [37] | $\checkmark$ | $\times$ | $\times$ | $\times$ | $\times$ | $\checkmark$ | PLs are class-dependent |
| PiCO [34] | $\checkmark$ | $\times$ | $\checkmark$ | $\checkmark$ | $\underline{\checkmark}$ | $\checkmark$ | Same class representations cluster |
| DPLL [38] | $\checkmark$ | $\times$ | $\checkmark$ | $\times$ | $\underline{\checkmark}$ | $\times$ | Input is invarient to the translations |
| SoLar [33] | $\checkmark$ | $\checkmark$ | $\checkmark$ | $\checkmark$ | $\checkmark$ | $\times$ | High-confid. sample is likely correct |
| PaPi [40] | $\checkmark$ | $\checkmark$ | $\checkmark$ | $\checkmark$ | $\checkmark$ | $\underline{\checkmark}$ | Same class representations cluster |
| CroSel [30] | $\checkmark$ | $\checkmark$ | $\checkmark$ | $\checkmark$ | $\checkmark$ | $\underline{\checkmark}$ | Stable high-confid. sample is correct |

when these additional techniques result in significant improvements, they tend to boost the model's ability to learn representations rather than resolving the inherent ambiguities of PLs.

The main contributions of our paper can be summarized as follows: (i) We advance the understanding of PLL taxonomy and establish minimal algorithm design principles. At the core of these principles is mini-batch PL purification, a fundamental aspect that goes beyond using supervised information. These principles not only enhance the efficiency of algorithm development but also act as a conclusive work to prevent redundant efforts in future research. (ii) We analyze the design philosophies and component frameworks of SOTA PLL methods, conducting comprehensive studies on benchmark-simulated datasets with PLs. Building on this, we highlight a *minimal working algorithm* that adheres to our design principles, and propose an enhancement strategy to mini-batch PL purification that have the potential to elevate performance across all existing SOTA methods.

## 2 Preliminaries

### 2.1 Notation

Consider a $k$-class classification problem. Let $\boldsymbol{x} \in \mathcal{X}$ be features and $y \in \mathcal{Y} \doteq \{1, 2, \ldots, k\}$ be labels. Then one has $(\boldsymbol{x}, y)$ sampled from the ground-truth joint density $p(\boldsymbol{x}, y)$ over $\mathcal{X} \times \mathcal{Y}$ in supervised learning. PLL deals with PLL data $(\boldsymbol{x}, S)$, which is independently drawn from a corrupted distribution $p(\boldsymbol{x}, S)$ of $p(\boldsymbol{x}, y)$ with $p(\boldsymbol{x})$ unchanged. $S \in \{2^{[k]} \backslash \emptyset \backslash [k]\}$ denotes a PL, and $\mathcal{D} = (\boldsymbol{x}_i, S_i)_{i=1}^n$ is a PLL dataset. The key assumption of PLL is that the latent true label of an instance is always included in its PL, i.e., $p(y \in S | \boldsymbol{x}, S) = 1$. Let $\Delta^{k-1} \subset [0, 1]^k$ denote the $k$-dimensional simplex. Let $f : \mathcal{X} \to \Delta^{k-1}$ be a multi-label classifier to be trained, specifically, a composite of a backbone (e.g., ResNet [18]) and an inverse link function $\psi^{-1}$ [27] (e.g., softmax), so that $f(\boldsymbol{x})$ can be interpreted as probabilities. Let $\ell : \Delta^{k-1} \times \mathcal{Y} \to \mathbb{R}_+$ be a surrogate loss function, e.g., cross-entropy loss. The classification risk of $f$ is defined as $\mathcal{R}(f) = \mathbb{E}_{p(\boldsymbol{x}, y)}[\ell(f(\boldsymbol{x}), y)]$, which is the performance measure we would like to optimize.

### 2.2 Backgrounds

In this section, we review recent prominent PLL works. In PLL, a common practice is to adopt a weighted objective of the form

$$\mathcal{R}(f) = \mathbb{E}_{p(\boldsymbol{x}, S)}\Big[\sum_{z \in S} w(\boldsymbol{x}, z)\ell(f(\boldsymbol{x}), z)\Big], \tag{1}$$

with the optimal weights $w(\boldsymbol{x}, z) = p(z|\boldsymbol{x})$. If $\ell$ is the cross-entropy loss, the weight can be integrated directly into the loss function, acting as optimization target (pseudo-label) for $\boldsymbol{x}$ directly. In classical taxonomy for PLL [47], IBS trains predictive models based on fixed weights assigned to the training instances, i.e., $\hat{w}(\boldsymbol{x}, z) = \{0, 1\}^k$, while ABS sets uniform weights during training, i.e., $\hat{w}(\boldsymbol{x}, z) = 1/|S|$.

**Definition 2.1** (Mini-batch PL purification). Mini-batch PL purification is a process where for each mini-batch $\mathcal{B} \subset \mathcal{D}$ selected at iteration $t$, the weights are updated such that the distinction among candidate labels' contributions increases over iterations:

$$w_{t+1}(\boldsymbol{x}; f, S) = g(\text{model's confidence for } \boldsymbol{x} \text{ based on current and previous iterations}), \quad (2)$$

with $g$ being a strictly increasing function that increases the weight for more likely candidate labels according to the model's confidence. The model's parameters $\theta_t$ are updated by optimizing a weighted loss over $\mathcal{B}$:

$$\theta_{t+1} = \theta_t - \eta_t \nabla_\theta \sum\nolimits_{(\boldsymbol{x}, S) \in \mathcal{B}} \ell(f(\boldsymbol{x}; \theta_t), S; w_{t+1}(\boldsymbol{x})). \quad (3)$$

The standard practice is initializing the weights uniformly $w_i^0 = 1/|S|$ if $i \in S$ and $w_i^0 = 0$ otherwise, and let $f^0$ be initialized randomly. The model $f^0$ is then updated for at least one epoch to perform a preliminary training phase. Then in each mini-batch of $t$-epoch, $w(\boldsymbol{x})$ is computed where $f$ is fixed, and then $f$ is updated by the weighted objective where $w$ keeps fixed in backpropagation. An instantiation of mini-batch PL purification was first introduced by [25]. They use a delayed mechanism, i.e., $w(\boldsymbol{x})$ is computed by the output of historical model $f^{t-1}$ on $\boldsymbol{x}$, implicitly assuming that DNNs learn pattern first [2], and the delayed mechanism mitigates the accumulation of errors. Then, some methods replace the delayed mechanism with Match techniques to estimate $w$, which may rely on either Siamese networks [4] applied to two or more inputs (dual-augmentation match, i.e., DA), or co-teaching networks [16] where one network's outputs serve as targets for the other (dual-model match, i.e., DM). Furthermore, some methods advocate DA+DM framework; for example, PiCO [34] involves mutual guidance of two heterogeneous classifiers, with one built on top of a supervised contrastive learning architecture [20]. In addition techniques borrowed from various communities have propelled PLL methods to top performance, such as mixup [46] and data augmentation like cropping and flipping [28], which have become mainstream.

As shown in Table 1, prominent methods in recent years rely on the mini-batch PL purification strategy and specific enhanced tools. However, our research reveals a significant disparity in the impact of these components, at least on current benchmarks. Mini-batch PL purification is sufficient to provide a reliable guarantee of performance, while the additional tools contribute relatively little. Although techniques like data augmentation can enhance the performance by improving the model's robustness to input variations, they primarily boost representation learning, but not benefits the disambiguition for PLL.

## 3 Understanding Minimal Algorithm Design Principles

In this section, we inverstigate four SOTA PLL methods that have consistently demonstrated top accuracy across various benchmark tasks. By methodically dissecting these methods and analyzing the components credited for their robustness against PLs, we distill the essential elements contributing to their success. We defer the experiments details to Appendix.

### 3.1 PLL with DA Match

**Algorithm details.** The key contribution of DPLL [38] lies in incorporating neighborhood consistency, a technique adapted from semi-supervised learning, into PLL. This technique maximizes the similarity among several perturbed views of the same instance, thereby inducing smoothness in the structure of learned representations, referred to as dual-augmentation match (DA Match).

Specifically, DA Match instantiates Eq. 1 by specifying the loss function as the KL divergence and estimating the weights by a weighted sum of outputs of all augmentations in $\mathcal{A}(\boldsymbol{x})$:

$$\mathcal{R}(\boldsymbol{x}, f) = -\sum\nolimits_{\mathcal{A}(\boldsymbol{x})} \hat{w}(\boldsymbol{x})^\top \log f(\mathcal{A}(\boldsymbol{x})), \quad (4)$$

$$\hat{w}(\boldsymbol{x}, z) = \begin{cases} \dfrac{\left(\prod_{\alpha \in \mathcal{A}(x)} f_z(\alpha)\right)^{1/|\mathcal{A}(x)|}}{\sum_{j \in S}\left(\prod_{\alpha \in \mathcal{A}(x)} f_j(\alpha)\right)^{1/|\mathcal{A}(x)|}} & z \in S, \\ 0 & \text{otherwise.} \end{cases} \quad (5)$$

$\hat{w}$ is updated in a mini-batch-wise manner along with the model parameters and is initialized uniformly (mini-batch PL purification). In addition, the learning objective of DPLL includes another loss function as

$$\ell(\boldsymbol{x}, f) = -\sum\nolimits_{i \notin S} \log(1 - f_i(\mathcal{A}(\boldsymbol{x}))), \quad (6)$$

where the vector subscript indicates the element is that position. It encourages the output of each non-candidate label to be zero.

**Evaluation 1. Specific implementations do not matter.**    First, we replace the two terms in learning objective with alternative approaches. Eq. 6 is conceptually equivalent to encouraging the sum of the outputs for the candidate labels to be close to one, i.e,

$$\ell(\boldsymbol{x}, f) = -\log \sum_{i \in S} f_i(\mathcal{A}(\boldsymbol{x})). \tag{7}$$

Instead of using a shared target for all views, we modifies the optimization target for each view separately based on the output of the other view:

$$\mathcal{R}(\boldsymbol{x}, f) = -\big(\hat{w}_2(\boldsymbol{x})^\top \log f(\boldsymbol{x}_1) + \hat{w}_1(\boldsymbol{x})^\top \log f(\boldsymbol{x}_2)\big), \tag{8}$$

where $\boldsymbol{x}_1, \boldsymbol{x}_2 \in \mathcal{A}(\boldsymbol{x})$, and

$$\hat{w}_{2(1)}(\boldsymbol{x}, z) = \begin{cases} f_z(\boldsymbol{x}_{1(2)})/\sum_{j \in S} f_j(\boldsymbol{x}_{1(2)}) & z \in S, \\ 0 & \text{otherwise.} \end{cases} \tag{9}$$

We combine two DA Match terms and two loss functions in pairs, respectively. As can be seen from Table 2, there was no significant difference between the results of these four combinations. This indicates that the specific implementation methods of DA Match, including the exact form of the loss function, is not critical.

**Evaluation 2. Additional losses do not matter.**    We then split the combined learning objectives into two separate objectives to examine the difference in the contribution of these two components to learning. We found that Eq. 6 and Eq. 8 were comparable under the relatively simple settings, but Eq. 8 outperformed Eq. 4 in challenging scenarios (CIFAR-100 and mini-ImageNet). Note that Eq.6 is a traditionally considered ABS loss, and Eq. 8 implements mini-batch PL purification. It is commonly believed that ABS does not require identifying the true labels during training, leading to over-parameterized DNNs memorizing all candidate labels [44], which results in poor performance. However, as discussed in Section 1, ABS can still achieve acceptable performance because the optimization process may induce differentiated outputs. We will elaborate on this in more detail in Section 3.4. Another noteworthy observation is that Eq. 8 not only did not lead to a decrease in accuracy compared to the original DPLL, but even showed some improvement, which is likely because Eq. 8 can model neighborhood consistency better than Eq. 4.

So far, we have extracted a core unit from DPLL, which takes the form of Siamese networks [4]: a weight-sharing network applied on two (or more) inputs for comparing, and PLs prevent the model from collapsing, i.e., outputting a constant for all inputs. We call Eq. 8 *dual augmentations single model* (DASM), as depicted in Figure 5. However, it remains to be seen whether the effectiveness of DASM is due to mini-batch PL purification or neighborhood consistency.

**Evaluation 3. Mini-batch PL purification does matter.**    We modify the learning strategy of DASM by either altering the mini-batch PL purification strategy or removing the consistency component. We compare them in Table 2, where the "-H", "-S" or "-E" suffix means using the hard pseudo-labels, one-step iteration or epoch-wise iteration. Specifically, DASM-H uses hard labels as optimization targets for Eq. 8:

$$\hat{w}_2(\boldsymbol{x}) = \boldsymbol{e}^i, \hat{w}_1(\boldsymbol{x}) = \boldsymbol{e}^j, \text{ where } i = \arg\max f(\boldsymbol{x}_1) \text{ and } j = \arg\max f(\boldsymbol{x}_2), \tag{10}$$

where $\boldsymbol{e}^i$ denotes the $i$-th standard canonical vector, i.e., $\boldsymbol{e}^i \in \{0, 1\}^k$, $\mathbf{1}^\top \boldsymbol{e}^i = 1$. DASM-S simulates one-step EM methods by training the model with uniform targets for the first 50 epochs, and then transforming PLL into supervised learning by using one-hot pseudo-labels (i.e., the `argmax` of the model's output for each instance at the 50th epoch) for the next 450 epochs. The results indicated that the performance of DASM-H and DASM-S was inferior to DASM. This can be attributed to the models' inability to adjust learning targets at the *appropriate* time based on the underlying learned patterns: Since DNNs tend to fit easy patterns first and gradually memorize harder ones, a phenomenon known as memorization effects [2], DASM-H may remember unreliable information due to random initialization, and DASM-S may lead remember too much undesired memorization [15]. DASM-E shifts from a mini-batch-wise manner to an epoch-wise manner, performing pseudo-label estimation and model updates at the epoch level, which resulted in decreased accuracy. Compared to DASM-E, mini-batch manner benefits from using the up-to-date model for generating optimization objectives

Table 2: Conceptual and empirical comparisons (%) of various simplifications of DPLL. ✓/✗ indicates whether a technique is used.

| Methods | Mini-b. purif. | ABS loss | Data augment. | DA | FMNIST 0.3 | FMNIST 0.7 | CIFAR-100 0.05 | CIFAR-100 0.1 | mini-I.Net ins.-dep. |
|---|---|---|---|---|---|---|---|---|---|
| Eq. 4+6 DPLL | ✓ | ✓ | ✓ | ✓ | 93.82 | **92.68** | 76.81 | 75.93 | 52.22 |
| Eq. 4+7 | ✓ | ✓ | ✓ | ✓ | 93.80 | 92.44 | 79.35 | 78.85 | 53.40 |
| Eq. 8+6 | ✓ | ✓ | ✓ | ✓ | 93.49 | 92.19 | **79.75** | 78.87 | 53.78 |
| Eq. 8+7 | ✓ | ✓ | ✓ | ✓ | 93.57 | 92.20 | 79.53 | 78.85 | 54.09 |
| Eq. 6 | ✗ | ✓ | ✓ | ✗ | 93.58 | 92.12 | 76.96 | 75.94 | 44.69 |
| Eq. 8 DASM | ✓ | ✗ | ✓ | ✓ | 93.89 | 92.85 | 79.70 | **79.62** | 54.71 |
| DASM-H | ✗ | ✗ | ✓ | ✓ | 93.86 | 92.37 | 78.25 | 33.22 | 34.59 |
| DASM-S | ✗ | ✗ | ✓ | ✓ | 93.28 | 90.75 | 78.65 | 76.22 | 36.71 |
| DASM-E | ✗ | ✗ | ✓ | ✗ | 93.80 | 92.35 | 79.30 | 79.11 | 53.44 |
| SASM | ✓ | ✗ | ✓ | ✗ | **93.83** | 92.18 | 79.38 | 78.19 | **55.45** |
| DASM w/o.aug | ✓ | ✗ | ✗ | ✓ | 90.86 | 89.60 | 60.18 | 56.39 | 30.85 |

and also improves computational efficiency. In addition, we change the dual feedforwarding to single. The optimization target of an input is modified in place according to its own output, which we term *single augmentations single model* (SASM). It was somewhat surprising that this method does not require additional components, performed remarkably well, implying DA may not be necessary.

**Evaluation 4. Does data augmentation matter?**    Additionally, removing data augmentation from DASM, as in DASM w/o.aug, causes a decrease in accuracy as expected. We will discuss its impact further in Section 3.3.

## 3.2   PLL with DA+DM Match

**Algorithm details.**    PiCO [34] enhances representation learning by incorporating supervised contrastive learning into PLL. It utilizes two heterogeneous classifiers sharing a backbone (one linear and one contrastive-based), guiding each other to instantiate Eq. 1. PaPi [40] investigates PiCO and identifies limitations in the contrastive learning module. Thus PaPi adopts a more efficient approach inspired by the delayed mechanism in PRODEN, instantiating Eq. 1 without the need to maintain two separate models. Besides, PaPi also uses the zero-and-normalized outputs of historical models to guide a prototypical classifier that shares the same backbone with the linear classifier. Both methods feed forward different views of the same input. We refer to such franework as dual-augmentation and dual-model match (DA+DM Match). CroSel [30] is the latest PLL method that, in addition to using the DM framework to generate optimization targets for each other, also selects samples with more accurate pseudo-labels for the other model to compute supervised loss.

**Evaluation.**    At a high level, PiCO draws inspiration from co-teaching [16]: instead of training a single classifier, it trains two classifiers simultaneously and lets them teach each other in every mini-batch. We simplify this idea by removing the contrastive-based classifier and using two networks with the same architecture but different initialization, which is also CroSel without its sample selection module. We call such method *dual augmentations dual models* (DADM). Taking it a step further, *single augmentation dual models* (SADM) removes one data augmentation, feeding both networks the same view of an instance within the same epoch. Conversely, if we cancel DADM from CroSel, the remaining implementation is co-teaching adapted for PLL (Coteaching in Tabel 4). For PaPi, we strip away the prototypical classifier, resulting in a streamlined version akin to PRODEN with added data augmentation (PRODEN+), to explore whether the specific instantiation of DM makes a difference. Figure 5 illustrates their basic workflow.

Our results are shown in Table 4. Both DADM and SADM outperformed PiCO and were comparable with CroSel, and PRODEN+ generally matches the performance of PaPi. Deleting the implementation of mini-batch PL purification, whether by altering the iteration frequency or replacing soft pseudo-

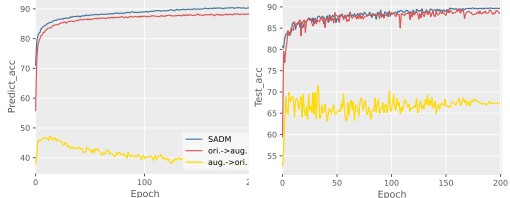

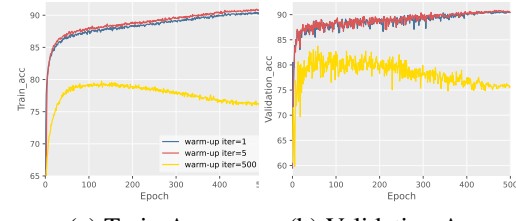

(a) Acc of pseudo-labels    (b) Test Acc

(a) Train Acc    (b) Validation Acc

Figure 2: Comparative results of different training setups of DASM on FMNIST with PLs. (a) shows the pseudo-label accuracy, while (b) presents the test accuracy.

Figure 3: Warm up different iterations on FM-NIST with PLs. Red lines mean terminating the warm-up phase at a local maximum in validation accuracy.

labels with hard pseudo-labels, consistently leads to a decline in performance. The observations reaffirm that mini-batch PL purification is essential for achieving top performance. By comparing the performance of the DM framework with the SM framework, we discover that the DM framework's edge often comes from the diverse capabilities of the two networks, which help handle the noise introduced by PLs. On the other hand, the DA methods without any special design (DASM, DADM, PRODEN+) showed suboptimal performance than the SA methods, perhaps hinting at the importance of exploring different augmentation methods [31] to discover appropriate choices for class-invariant patterns, remaining for future research.

### 3.3 Does Data Augmentation Help Identification Better?

Our findings indicate that mini-batch PL purification and data augmentation are pivotal for PLL, achieving competitive performance when both techniques are implemented. Data augmentation is a well-established regularization tool, enhancing model robustness to input variations. In the following, we explore whether its role in PLL extends beyond this conventional purpose, specifically whether it facilitates the crucial task of identifying the true label among candidate labels. If data augmentation aids true-label identification, we would expect that augmented examples generate more accurate pseudo-labels, then enhancing the performance. Specifically, we experiment with DASM by setting one view as augmented and the other as non-augmented. We also use different training setups: (i) using zero-and-normalized pseudo-labels generated from original $x$ to supervise augmented $x$, and (ii) the reverse, using augmented $x$ to supervise original $x$. The results are presented in Figure 2.

Switching from dual paths to a single path while retaining supervised learning on augmented instances with pseudo-labels from the original instances showed little change in accuracy. However, when using pseudo-labels from augmented instance to supervise original one, performance drops significantly. This suggests that data augmentation alone is insufficient to preserve the mutual information between examples and their true labels, as augmented views may discard task-relevant information, thereby degrading performance. Following this reasoning, our results suggest that data augmentation indirectly benefits the classifiers built upon representation learning rather than aiding in label disambiguation. Therefore, it should NOT be considered a design principle for PLL.

### 3.4 PLL through Pseudo-Label Manipulation vs. Loss Minimization

Mainstream methods mentioned above conduct mini-batch PL purification primarily by directly manipulating the optimization targets of candidate labels. However, as analyzed in Section 1 and observed in Eq. 6 of Table 2, using stochastic algorithms to minimize loss functions, even ABS loss funtions that are traditionally considered to "treat all candidate labels equally", can also implicitly result in a progressive effect of pseudo-labels. To our knowledge, no existing research has explored the relationship between pseudo-label manipulation and loss minimization. Our findings raise the question: how do these two methods compare in terms of their effectiveness and mechanisms in the context of PL identification?

We investigate several representative PLL losses, which are traditionally thought as ABS:
- Modified negative log likelihood loss (Eq. 6) [38]: $\ell_{\text{neg}}(\boldsymbol{x}, f) = -\sum_{i \notin S} \log(1 - f_i(\boldsymbol{x}))$;
- APL loss [24]: $\ell_{\text{APL}}(\boldsymbol{x}, f) = \frac{1}{|S|} \sum_{i \in S} \tilde{\ell}(f(\boldsymbol{x}), i)$, with GCE loss [49] as the component $\tilde{\ell}$;

- Maximum likelihood loss [19]: $\ell_{\mathrm{maxi}}(\boldsymbol{x}, f) = -\log \sum_{i \in S} f_i(\boldsymbol{x})$.

Denote the scores of $\boldsymbol{x}$ outputted by the last layer before `softmax` as $\boldsymbol{z}$, i.e., $\psi^{-1}(\boldsymbol{z}) = f(\boldsymbol{x})$ where $f = (\psi^{-1} \circ f^{(n)} \circ \cdots \circ f^{(1)})$. Let us look at the gradients of $\ell_{\mathrm{neg}}$ and $\ell_{\mathrm{APL}}$:

$$
\frac{\partial \ell_{\mathrm{neg}}}{\partial \boldsymbol{z}_k} = 
\begin{cases}
-f_k \sum_{i \notin S} \frac{e^{\boldsymbol{z}_i}}{\sum_{j \in \mathcal{Y}} e^{\boldsymbol{z}_j - e^{\boldsymbol{z}_i}}} & k \in S, \\
f_k (1 - \sum_{i \notin S, i \neq k} \frac{e^{\boldsymbol{z}_i}}{\sum_{j \in \mathcal{Y}} e^{\boldsymbol{z}_j - e^{\boldsymbol{z}_i}}}) & \text{otherwise,}
\end{cases}
\tag{11}
$$

$$
\frac{\partial \ell_{\mathrm{APL}}}{\partial \boldsymbol{z}_k} = 
\begin{cases}
-\frac{1}{|S|}(f_k \sum_{i \in S, i \neq k} f_i^q - f_k^q(1 - f_k)) & k \in S, \\
\frac{1}{|S|} f_k \sum_{i \in S} f_i^q & \text{otherwise,}
\end{cases}
\tag{12}
$$

where $q \in (0, 1]$ is a tunable parameter of GCE. From a gradient perspective, we observe that while the implicit optimization targets for all candidate labels are the same for each loss function, their optimization speeds differ. Specifically, the gradients of the candidate labels are consistently negative until one candidate label accumulates all the probabilities, which implies that both loss functions promote the output of each candidate label to converge to 1. As a result, the candidate labels compete for dominance. A label with a higher output probability experiences a larger gradient and thus becomes the winner. According to the memorization effects of DNNs, such a label is more likely to be the true label receiving a larger gradient at the beginning, which explains why the labels become distinguishable with these two losses. However, since the optimization targets for all candidate labels remain the same, minimizing these two losses does not align with our definition of mini-batch PL purification where the optimization targets are expected to diverge progressively.

Then we focus on the third loss $\ell_{\mathrm{maxi}}$. We examine its gradients as

$$
\frac{\partial \ell_{\mathrm{maxi}}}{\partial \boldsymbol{z}_k} = 
\begin{cases}
f_k - \frac{e^{\boldsymbol{z}_k}}{\sum_{i \in S} e^{\boldsymbol{z}_i}} & k \in S, \\
f_k & \text{otherwise.}
\end{cases}
\tag{13}
$$

It is crucial to recognize two key points: (i) The philosophy of $\ell_{\mathrm{maxi}}$ is to ensure that the output of all candidate labels sums to 1, which is *exactly the same* as that of $\ell_{\mathrm{neg}}$, which requires the output of all non-candidates to sum to 0. However, their gradients are *completely different*, indicating that they lead to different optimal empirical solutions even with identical initialization and stochastic optimizer. (ii) When learning with $\ell_{\mathrm{maxi}}$, the implicit optimization target of an instance is the zero-and-normalized outputs of current model on this instance itself, exhibiting *consistent behavior with pseudo-label manipulation* in SASM.

**Lemma 3.1.** *Suppose using the same stochastic optimizer, then performing SASM is mathematically equivalent to minimizing $\ell_{\mathrm{maxi}}$.*

Empirical results in Table 5 also verified the theoretical findings. However, pseudo-label manipulation offers greater flexibility as it allows for more arbitrary modifications to the optimization targets, as DASM, SADM, etc. have done, and additional steps over generating soft pseudo-labels, such as sharpening [29], but the targets of loss functions are fixed.

Now, our empirical investigations have firmly established the necessity of mini-batch PL purification along with using supervision. Supervision can be used in two ways: directly within the loss function as it in supervised learning, or to manipulate pseudo-labels on-the-fly. We identify SASM is the minimal working algorithm. It achieved superior or at least comparable performance compared with SOTA PLL methods in most cases, while not requiring multiple forward propagations or additional components.

### 3.5 Probing the Implementation of Mini-Batch PL Purification

Here, we examine the implementation of mini-batch PL purification, raising at least two questions: Q1. Does different initialization methods impact the performance? Q2. Why does model confidence in PLs is effective to disambiguation?

In early training stages, our primary concern is if initialization methods might cause certain candidate labels to receive significantly higher confidence. Consider a neural network where weights are independently drawn from a standard normal distribution with zero mean and variance $\sigma^2$ (as in normal initialization, Xavier initialization [1], He initialization [17]). For an instance $\boldsymbol{x}$, the output $z$ of a neuron after a ReLU activation is given by $z = \max(0, \boldsymbol{w}\boldsymbol{x} + b)$, where $\boldsymbol{w}$ represents the

weight vector and $b$ is a small constant bias. By the Central Limit Theorem, the mean of the weights $\bar{w} = 1/n \sum_{i=1}^{n} w_i$ for a sufficiently large number of neurons $n$ approaches a normal distribution with mean zero and variance $\delta^2/n$. Applying Chebyshev's inequality, we have $P(|\bar{w}| \geq a) \leq \delta^2/(na^2)$. As $n \to \infty$, $\bar{w}$ is close to zero with high probability. Then the output for each neuron $z$ will be close to $b$. By a similar reasoning, this result generalizes to deeper layers, suggesting that the initial outputs across classes approximate a uniform distribution regardless of the number of layers.

**Lemma 3.2.** *For a neural network initialized with weights $w$ drawn from a normal distribution $\mathcal{N}(0, \delta^2)$, the initial outputs across classes approximate a uniform distribution as the number of neurons $n \to \infty$.*

Then we explain why using model's output as a proxy for the probability that a candidate label is the true label works. This practice of using the high-confidence label as the true label, introduced to PLL by [25], has long been foundational in noisy-label learning (e.g., [16, 23]). Its success in PLL hinges on a key assumption: the true label has a higher probability of appearing among the candidate labels than any incorrect label. In fact, the experimental setups employed across PLL methods impose a further restriction on this assumption, that is, $p(y \in S|x, S) = 1, p(i \in S|x, S) > 1, \forall i \neq y$. This setup implies that, within any sufficiently small neighborhood in the data space, the true label will dominate. Consequently, when stochastic gradient-based optimizers are used, the true label tends to contribute more frequently to the objective function, making it more likely to be learned first.

## 4 Improving Mini-Batch PL Purification

### 4.1 Motivation

The standard practice involves initiating the process with uniform pseudo-labels for one epoch to bootstrap the classifier's basic capabilities. Our first observation is that the effect of warming up for one epoch is nearly indistinguishable from not warming up at all, prompting further investigation into the actual efficacy of the warm-up phase. Suppose a neural network is initialized with weights drawn from a Gaussian distribution with mean zero and a sufficiently small variance. If the inputs are normalized, the pre-activation values across a neural network tend to be small and centered around zero. When these values are input to a softmax function, the resulting distribution across classes tends to uniformity.

Our second observation is that a prolonged warm-up using uniform pseudo-labels often leads to the network overfitting to candidate labels, as illustrated in Figure 3, evident around 50 epochs with a decline in validation accuracy. While overfitting to the training data becomes apparent about 120 epochs, marked by a drop in training accuracy after previously reaching a peak.

We simply terminate the warm-up phase at a local maximum in validation accuracy, about 5 epoch, preventing excessive memorization while preserving more information beneficial for generalization. Then the results showed a little improvement. Several approaches [34, 24] have used multiple epochs for warm-up and treat the number of warm-up epochs as a hyperparameter. However, due to the varying difficulty levels across samples, using a uniform duration for the warm-up phase could result in performance disparities among different subgroups within the dataset. This suggests the need for an adaptive warm-up strategy.

### 4.2 StreamPurify: An Instance-Dependent Warm-Up Strategy

Building upon the analysis and our established design principles, we propose StreamPurify, a novel instance-dependent warm-up strategy, which fine-tunes the entry into the mini-batch PL purification phase based on each instance's readiness. During the initial training phase, it selectively channels instances that have higher confidence in the accuracy of their pseudo-labels into the PL purification phase, while others continue training with uniform targets until they meet the readiness criteria. This filtered progression helps prevent DNNs from harmfully memorizing incorrect pseudo-labels and mitigate the accumulation of errors from inadequately learned samples. Differing from sample selection techniques used in noisy-label learning [36] that reevaluated samples in every iteration, our filter approach is conducted without replacement. Once the training samples are transitioned to the purification phase, they do not revert to uniform targets, ensuring that the strategy remains in line with mini-batch PL purification.

Table 3: Classification accuracy (%) improvement of PLL methods with StreamPurify.

| Methods | FMNIST | | CIFAR-10 | | CIFAR-100 | | mini-I.Net |
| | 0.3 | 0.7 | 0.3 | 0.7 | 0.05 | 0.1 | ins.-dep. |
| --- | --- | --- | --- | --- | --- | --- | --- |
| SASM | 0.15 | 0.09 | 0.02 | 0.34 | 0.30 | 0.21 | 1.82 |
| SADM | 0.05 | 0.22 | 0.10 | 0.18 | 0.19 | 0.58 | 0.99 |
| DADM | 0.03 | 0.02 | 0.03 | 0.46 | 0.66 | 0.51 | 0.72 |
| DASM | 0.36 | 0.26 | 0.13 | 0.29 | 0.22 | 0.42 | 1.41 |
| DPLL | 0.11 | 0.00 | 0.06 | 0.40 | 0.85 | 0.47 | 0.95 |
| PiCO | 0.29 | 0.43 | 0.60 | 0.49 | 0.35 | 0.62 | 1.03 |
| PaPi | 0.17 | 0.36 | 0.30 | 0.20 | 0.68 | 0.47 | 0.89 |

StreamPurify is compatible to existing PLL methods and can absorb various sample selection criteria, such as small-loss trick. We adopt the sample selection method from CroSel [30] that chooses the samples with stable and high confidence as a filter criterion within StreamPurify, and combine it with the mini-batch PL purification approaches discussed in our paper. Empirical evaluations indicate that methods augmented with StreamPurify generally exceed the performance of their conventional counterparts, validating the effectiveness of StreamPurify, especially in complicated learning scenarios. This substantiates the robustness and adaptability of instance-dependent warm-up, suggesting it as a promising direction for future improvements in PLL.

## 5 Discussion and Future Work

We have systematically delineated the core components underlying successful PLL methods, centering our insights around the pivotal effect of mini-batch PL purification. We have proposed StreamPurify, an enhanced mini-batch PL purification strategy that tailors the learning path for each sample based on its state of readiness. We reaffirm that strictly categorizing PLL methods as IBS or ABS may oversimplify the dynamics of how these methods operate in practice. In the future, we wish to explore advanced PLL approaches guided by the minimal algorithm design principles. We also wish to extend the applicability and understanding of PLL methods to domains beyond vision tasks.

## Acknowledgements

This research was supported by the National Science Foundation of China (62406066, 62206050, 62125602, 62176055, 62076063), Jiangsu Province Science Foundation for Youths (BK20241297, BK20210220), China Postdoctoral Science Foundation (2021M700023), Young Elite Scientists Sponsorship Program of Jiangsu Association for Science and Technology (TJ-2022-078), the Australian Research Council Discovery Early Career Research Award (DE230101116), the Fundamental Research Funds for the Central Universities (2242024k30035), the Big Data Computing Center of Southeast University, JST CREST Grant Number JPMJCR18A2 and a grant from Apple, Inc. Any views, opinions, findings, and conclusions or recommendations expressed in this material are those of the authors and should not be interpreted as reflecting the views, policies or position, either expressed or implied, of Apple Inc.

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

# Appendix

## A    Experiments Details

As benchmarking on partially labeled vision datasets has become standard practice in evaluating deep PLL methods, we conduct experiments on FMNIST [41], CIFAR-100 [21] and mini-ImageNet [32]. We generated PLs by the Flipping strategy [24] for FMNIST, CIFAR-10 and CIFAR-100. Each label $i$ is added into PL with a flipping probability $\eta_i^y = p(i \in S|y)$ independently and features are untouched: $p(s|x,y) = M \prod_{i \in S} \eta_i^y \prod_{i \notin S} (1 - \eta_i^y)$ where $M = 1/\left(1 - \prod_{i \neq y} \eta_i^y\right)$. We assumed $\eta_i^y = \eta, \forall i \neq y$ and $\eta_y^y = 1$, and set $\eta = \{0.3, 0.7\}$ on FMNIST and CIFAR-10, $\eta = \{0.05, 0.1\}$ on CIFAR-100. For mini-ImageNet and ablation experiments (Figure 1, 2, 3, 4), we simulated the real scenario by setting the flipping probability for each incorrect label individually for each instance. We first trained a classifier with clean labels, and then for each instance, set the confidence prediction of the classifier as the flipping probability [42].

Our explorations used three backbones: 5-layer LeNet [22] on FMNIST, 18-layer ResNet [18] on CIFAR-10 and CIFAR-100, and 34-layer ResNet [18] on mini-ImageNet. All the methods were trained for 500 epochs with a standard SGD optimizer [9] with a momentum of 0.9 and the batch size was 256 (128 for mini-ImageNet). We left out 10% of the corrupted training samples as a validation set, and searched the initial learning rate from $\{0.1, 0.07, 0.05, 0.03\}$ with cosine learning rate scheduling. We conducted 3 trials for each experiment, and recorded the mean test accuracy in percentage. There were two kinds of random augmentations involved. "Weak" augmentation was a random horizontal flips and crops [3]. For "strong" augmentation on FMNIST and CIFAR-10, we added Cutout [8] to the weak augmentation, and on CIFAR-100 and mini-ImageNet, we additional leveraged AutoAugment [7]. We denote the augmentation by $\mathcal{A}(\cdot)$, with method clear from context. The implementation was based on PyTorch [26] and experiments were carried out with GeForce RTX 4090 D.

Notably, we focused on the core components of the SOTA PLL methods mentioned in this paper, rather than strictly adhering to the settings detailed in their original implementations. This approach was taken to ensure fair and meaningful comparisons. For instance, we did not include techniques such as mixup in PaPi or the triple augmentation used in DPLL. As a result, the reported performance metrics in our paper might be slightly lower than those presented in the original publications.

## B    Experimental Results

In Figure 1 (c-d), CLPL is a traditionally considered ABS loss from [6]. Figure 4 illustrates the dynamic changes in the same method under different hyperparameters and optimization methods. One can expect that with more extreme hyperparameters or optimization methods, the approach may degrade to one-step EM or persist with nearly uniform optimization targets. This indicates that method categorization must be assessed on a case-by-case basis post hoc. Figure 5 is the workflow of several key methods proposed in the paper. Table 4 provides conceptual and empirical comparisons of various simplifications of PiCO, PaPi and CroSel. Consistent with Table 2, the results reiterate that mini-batch PL purification is pivotal to achieve top performance. Table 5 presents the results of minimizing three loss functions. It is evident that $\ell_{\text{maxi}}$ yield similar results with SASM, and owing to the mini-batch PL purification, and due to the implementation of mini-batch PL purification, the accuracy is higher than the other two losses.

Table 4: Conceptual and empirical comparisons (%) of various simplifications of PiCO, PaPi and CroSel.

| Methods | Mini-b. purif. | Data augment. | DA | DM | FMNIST 0.3 | FMNIST 0.7 | CIFAR-100 0.05 | CIFAR-100 0.1 | mini-I.Net ins.-dep. |
|---|---|---|---|---|---|---|---|---|---|
| PiCO | ✓ | ✓ | ✓ | ✓ | 93.40 | 91.64 | 76.11 | 75.65 | 48.36 |
| DADM | ✓ | ✓ | ✓ | ✓ | 93.59 | 92.40 | **80.28** | 79.13 | 53.69 |
| DADM-H | × | ✓ | ✓ | ✓ | 92.60 | 85.13 | 79.99 | 35.72 | 36.30 |
| DADM-S | × | ✓ | ✓ | ✓ | 92.12 | 87.81 | 78.16 | 74.66 | 36.42 |
| DADM-E | × | ✓ | ✓ | ✓ | 93.66 | 92.22 | 80.08 | 78.49 | 52.18 |
| SADM | ✓ | ✓ | × | ✓ | **94.02** | 92.36 | 80.11 | **79.65** | 54.64 |
| SADM-E | × | ✓ | × | ✓ | 93.72 | 92.51 | 89.85 | 78.66 | 53.08 |
| PaPi | ✓ | ✓ | ✓ | ✓ | 93.54 | 91.54 | 80.10 | 79.62 | **57.10** |
| PRODEN+ | ✓ | ✓ | ✓ | ✓ | 93.70 | **92.52** | 79.85 | 79.51 | 52.39 |
| PRODEN | ✓ | × | × | ✓ | 91.61 | 90.45 | 62.47 | 59.10 | 29.21 |
| CroSel | ✓ | ✓ | ✓ | ✓ | 93.84 | 92.31 | 80.40 | 80.06 | 53.58 |
| Coteaching | × | ✓ | ✓ | ✓ | 93.86 | 92.37 | 79.25 | 33.22 | 41.34 |

Table 5: Average training / test accuracy (%) of learning with three loss functions.

| | FMNIST | | | | CIFAR-100 | | | | mini-I.Net | |
|---|---|---|---|---|---|---|---|---|---|---|
| | 0.3 | | 0.7 | | 0.05 | | 0.1 | | ins.-dep. | |
| | training | test | training | test | training | test | training | test | training | test |
| $\ell_{\mathrm{neg}}$ | 94.98 | 92.28 | 92.05 | 91.29 | 89.66 | 74.91 | 86.35 | 74.20 | 30.40 | 28.04 |
| $\ell_{\mathrm{APL}}$ | 93.53 | 93.01 | 88.10 | 89.09 | 87.15 | 72.68 | 78.31 | 68.10 | 45.92 | 43.01 |
| $\ell_{\mathrm{maxi}}$ | **95.39** | **93.77** | **93.01** | **92.17** | **93.45** | **78.91** | **90.56** | **78.20** | **55.39** | **54.72** |

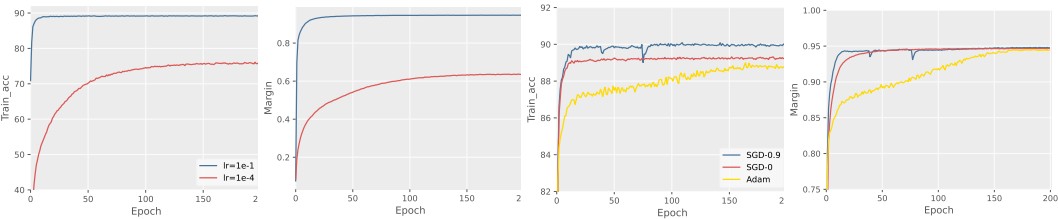

Figure 4: Training accuracy and confidence margin of predicted pseudo-labels for SASM with different learning rate or optimizer on FMNIST with PLs.

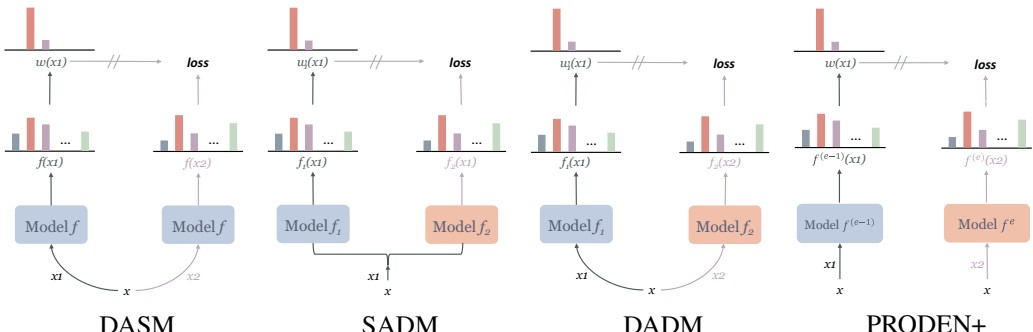

Figure 5: Illustrations of four effective units of SOTA PLL methods. We omit a symmetric loss from the other path, except PRODEN+ which can only compute one-path losses. "//" means stop gradient.

