# OpenReview forum: "What Makes Partial-Label Learning Algorithms Effective?"
_NeurIPS.cc/2024/Conference — NeurIPS 2024 poster_

### Official Review · Reviewer_b5oN · 2024-06-20

**Soundness:** 3
**Presentation:** 3
**Contribution:** 2
**Rating:** 6
**Confidence:** 4

**Summary:**

The paper examines the effectiveness of partial-label learning algorithms, identifying key factors contributing to their success. It discusses techniques from other fields that can enhance partial-label learning, such as the transition from uniform to one-hot pseudo-labels and the implementation of minimal algorithm design principles crucial for optimal performance. The document provides valuable insights into improving the performance of partial-label learning methods, making it a valuable resource for researchers in partial label learning.

**Strengths:**

1. Through a comprehensive empirical analysis, the paper uncovers key factors that drive the effectiveness of partial-label learning algorithms.
2. This paper is well-written.
3. The proposed warm-up strategy improved the state-of-the-arts performance.

**Weaknesses:**

1. The paper would benefit from providing additional details about the proposed warm-up strategy to enhance clarity and understanding.
2. Given the empirical nature of the paper, it is crucial to include detailed implementation information to ensure reproducibility and transparency.
3. The contribution of the baseline methods need to be mentioned in the paper.

**Questions:**

1. In Table 3, it is better to provide the accuracy of the SOTA methods with/without the proposed StreamPurify method.
2. In Table 3, it seems the improvements can be marginal. The performance improvement of most methods is within 1%.
3. More SOTA methods can be added to validate the effectiveness of the proposed method.

**Limitations:**

The paper lacks the discussion on the question of why this work has no societal impact or why the paper does not address societal impact.

---

> ### Author Rebuttal · Authors · 2024-08-07
>
> We would like to thank the reviewer for taking the time to review our work and the positive comments regarding its significance and writing quality. Below are our responses to the comments in Weaknesses and Questions.
>
> > C1. The paper would benefit from providing additional details about the proposed warm-up strategy to enhance clarity and understanding.
>
> Thank you for your suggestion! Our proposed StreamPurify is an instance-dependent warm-up strategy that can be compatible with various sample selection criteria. In our experiments, the selection criteria were a) the confidence of the highest predicted label exceeded a threshold (0.75 in our experiments), b) the label was within the candidate label set, and c) the highest confidence label remained stable during the first three epochs. When these three conditions were simultaneously met, we considered the sample ready to enter the PL purification process. The warm-up phase was concluded for all samples by the 10th epoch. We will include these details in the revised manuscript.
>
> > C2. Given the empirical nature of the paper, it is crucial to include detailed implementation information to ensure reproducibility and transparency.
>
> We have detailed all experimental setups in Appendix A. We will release the code and hyperparameters once the paper is accepted.
>
> > C3. The contributions of the baseline methods need to be mentioned in the paper.
>
> We have listed the techniques and main contributions of the baseline methods in Table 1 and highlighted their philosophical underpinnings and algorithmic details in Section 3. We would provide a more detailed explanation in the revised manuscript.
>
> > C4. In Table 3, it is better to provide the accuracy of the SOTA methods with/without the proposed StreamPurify method.
>
> Thank you for your suggestion. We will definitely include the detailed results in the revised version.
>
> > C5. Q1. In Table 3, it seems the improvements can be marginal. The performance improvement of most methods is within 1%. Q2. More SOTA methods can be added to validate the effectiveness of the proposed method.
>
> Thank you for your comments. There may be a misunderstanding regarding the main focus of our paper. Our primary contribution is understanding and uncovering the underlying principles of PLL methods. We identify that all SOTA methods conduct mini-batch PL purification, exhibiting a common behavior characterized by progressive transition from uniform to one-hot pseudo-labels.
>
> Building on this core contribution, we additionally observed that existing SOTA often overlook the importance of the initialization phase and tend to uniformly end the warm-up process for all samples before starting the PL purification process. However, samples’ learning progress varies at the beginning of training. Therefore, we propose a strategy that adaptively adjusts the timing for different samples to enter the PL purification process based on their readiness. The significance of our strategy lies more in its philosophy and concept rather than in the specific details of its implementation, making it flexible enough to be compatible with various sample selection criteria. In our paper, we used an existing method as a plug-in criterion and demonstrated the practical potential of StreamPurify on multiple datasets with different PL generation processes.
>
> In a word, **all the proposed methods in this paper primarily serve as demos to illustrate our findings rather than aiming to outperform SOTA methods**. Our strategy is expected to draw attention to the warm-up process, which can push forward PLL as a whole. We appreciate your suggestion and add two SOTA methods [1,2]. The test accuracy on FMNIST with instance-independent PLs is as follow:
>
> | method\flippingprobability | 0.3 | 0.7 |
> | --- | --- | --- |
> | [1] | 93.12 | 92.84 |
> | + StreamPurify | 93.67 | 93.41 |
> | [2] | 95.62 | 93.49 |
> | + StreamPurify | 95.70 | 94.19 |
>
> [1] PiCO+: Contrastive Label Disambiguation for Robust Partial Label Learning. TPAMI’23
>
> [2] Candidate-Aware Selective Disambiguation Based on Normalized Entropy for Instance-Dependent Partial-Label Learning. ICCV’23

---

> ### Author Response · Authors · 2024-08-12
> **Need further clarification?**
>
> Thanks very much for your constructive comments on our work.
>
> We have tried our best to address the concerns. Is there any unclear point so that we should/could further clarify?

---

> > ### Comment · Reviewer_b5oN · 2024-08-12
> >
> > Thank you for addressing my concern; I will raise my score.

---

> > > ### Author Response · Authors · 2024-08-14
> > >
> > > Thank you very much for dedicating your time to reviewing our paper and raisng the score!

---

### Official Review · Reviewer_RRwY · 2024-07-03

**Soundness:** 3
**Presentation:** 3
**Contribution:** 3
**Rating:** 6
**Confidence:** 4

**Summary:**

This paper presents an interesting survey of current popular PLL methods. The survey examines various aspects such as training techniques, optimization processes, and loss functions to identify the key factors that make PLL methods effective. Ultimately, the authors attribute the main reason for PLL's success to the mini-batch PL purification, which aligns with our intuition. Based on this finding, the authors propose an instance-dependent warm-up strategy StreamPurify, pointing the way for future research in PLL.

**Strengths:**

1. The analysis of the effectiveness of PLL methods is simple and clear.
2. The analysis from various perspectives is thorough.
3. The experiments are comprehensive.

**Weaknesses:**

1. The conclusion that "mini-batch PL purification is an important factor in making PLL methods effective" aligns with most current work and appears to be widely recognized.
2. The authors propose a method, StreamPurify, which seems similar to CroSel. Although the authors indicate that StreamPurify can use the sample selection method from CroSel, I believe this approach lacks novelty, as such methods have been widely used in many weakly supervised learning scenarios.
3. The paper seems to have not discussed different types of PL generation, which have been considered key assumptions in previous theoretical work, such as RC and CC. Different types of PL generation might lead to completely different results.

**Questions:**

1. The biggest challenge in PLL is considered to be the elimination of label ambiguity, specifically identifying the true label from candidate labels. This seems to be strongly correlated with mini-batch PL purification. Would observing label changes during training, as done in CroSel, potentially be a better way to identify the factors that make PLL methods effective?
2. For IBS and some compromise methods, is it true that once the model easily selects a label, it is less likely to change? Is this related to training settings such as the optimization method?
3. Can the expansion of the sample space brought by data augmentation be considered as helping contrastive methods to disambiguate labels, even though experiments suggest that data augmentation is not a key factor?
4. The focus of PLL work is on classification tasks. Are the observations in this paper still valid for regression tasks that have emerged recently?

**Limitations:**

N/A.

---

> ### Author Rebuttal · Authors · 2024-08-07
>
> We would like to thank the reviewer for taking the time to review our work and the positive comments regarding its significance. Below are our responses to the comments in Weaknesses and Questions.
>
> > C1. The conclusion that "mini-batch PL purification is an important factor in making PLL methods effective" aligns with most current work and appears to be widely recognized.
>
> Thank you for your insight. It is indeed a fact that mini-batch PL purification has become increasingly popular. A plausible explanation for this trend is that methods utilizing mini-batch PL purification yield better results, while traditional IBS and ABS are likely to perform poorly and thus were not be published in the literature. This observation just reaffirms the conclusion of our paper. However, it is important to highlight that existing studies **have not recognized** this factor as crucial to the effectiveness of PLL methods. Instead, they **typically focus on** adding more robust components, leading to SOTA methods quite complicated and different, and posing challenges in selecting the optimal direction for future algorithm design. Conclusive works like this paper should be vitally necessary.
>
> > C2. The authors propose a method, StreamPurify … I believe this approach lacks novelty, as such methods have been widely used in many weakly supervised learning scenarios.
>
> Thank you for the comment. There may be a misunderstanding regarding the concept and significance of StreamPurify. The main contribution of this paper --- the minimal algorithm design principles in PLL --- motivates us to observe that existing SOTA often overlook the initialization phase and tend to uniformly end the warm-up process for all samples before starting the PL purification process. However, samples’ learning progress varies at the beginning of training [1,2,3]. Therefore, we propose **a general strategy**, StreamPurify, which adaptively adjusts the timing for different samples to enter the PL purification process based on their readiness. The innovation of our strategy lies more in its philosophy, making it flexible enough to be compatible with various sample selection criteria. In our experiments, we used the criteria in CroSel as a plug-in method and demonstrated the practical potential of StreamPurify. A subtle yet important distinction between StreamPurify and sample selection methods like CroSel is that our strategy is conducted without replacement. This approach, driven by our philosophical motivation and minimal algorithm design principles, **has never been studied** in PLL community.
>
> We need to reiterate that all the proposed methods in this paper *primarily serve as demos* to illustrate our main understandings. We humbly believe that the significance and novelty of research should not be limited to algorithm design alone. Our findings and proposed strategy are expected to draw attention to the warm-up process, which can *push forward PLL as a whole*.
> Thank you again and we will provide a clearer discussion in the revised version.
>
> > C3. The paper seems to have not discussed different types of PL generation … Different types of PL generation might lead to completely different results.
>
> We have used different types of PL generation processes which are commonly used in the PLL community --- instance-independent PLLs and instance-dependent PLLs in our experiments. Furthermore, in our response to C2 of Reviewer ygK7, we added the class-dependent PL generation. Our findings are not influenced by different types of PL generation.
>
> > C4. Would observing label changes during training, as done in CroSel, potentially be a better way to identify the factors that make PLL methods effective?
>
> Thank you for raising this point, while upon reviewing the CroSel work, we did not find the definition and application of “label changes”. If you could provide more specific details, we would greatly appreciate it and would be happy to address any further questions. By the way, CroSel owes its advancement to “a cross selection strategy” and “select confident pseudo labels” in the original paper, both of which are analyzed through ablation experiments in Table 4, and the conclusions also support our claims.
>
> > C5. For IBS and some compromise methods, is it true that once the model easily selects a label, it is less likely to change? Is this related to training settings such as the optimization method?
>
> Thank you for asking. For IBS, as we mentioned in Section 1, traditional IBS methods have two stages: estimating pseudo-labels in the first stage and conducting supervised learning in the second stage. In this scenario, pseudo-labels are one-hot and remain fixed, and this is determined by the learning strategy and is not related to the optimization method. Sorry we are not clear on what is meant by “compromise methods”.
>
> > C6. Can the expansion of the sample space brought by data augmentation be considered as helping contrastive methods to disambiguate labels, even though experiments suggest that data augmentation is not a key factor?
>
> Thank you for the summary. Data augmentation is an integral part of training deep learning models with image data, enhancing the size and quality of training datasets [4]. Therefore, although we emphasize a novel point that data augmentation is not a key factor for addressing the PLL problem itself, since current PLL methods are validated on image datasets, data augmentation is a necessary part to boost final accuracy.

---

> ### Author Response · Authors · 2024-08-07
> **Response to Reviewer RRwY (2/2)**
>
> > C7. The focus of PLL work is on classification tasks. Are the observations in this paper still valid for regression tasks that have emerged recently?
>
> Thank you for raising this point. Since PL regression is not the focus of this paper, and is a relatively nascent area, we must be cautious and we cannot draw this conclusion.
>
> However, we can observe that SOTA PL regression methods [5] also follow mini-batch PL purification. Following the experimental setup in [5], we comparing our minimal working algorithm with SOTA PL regression methods IDent and PIDent on two datasets. The results of mean squared error are as follows. The results demonstrate that our method can effectively address PL regression problem.
>
> | dataset\method |  | IDent | PIDent | SASM |
> | --- | --- | --- | --- | --- |
> | Abalone | S=2 | 4.62 | **4.55** | 4.56 |
> | | S=4 | 4.66 | **4.58** | 4.63 |
> | | S=8 | 4.70 | 4.71 | **4.69** |
> | | S=16 | 4.90 | 4.90 | **4.84** |
> | Airfoil | S=2 | 15.58 | 14.99 | **14.23** |
> | | S=4 | 16.23 | 16.10 | **14.70** |
> | | S=8 | 17.81 |	17.86 | **16.04** |
> | | S=16 | 23.41 | 24.11 | **18.38** |
>
> [1] A Closer Look at Memorization in Deep Networks. ICML’17
>
> [2] FreeMatch: Self-Adaptive Thresholding for Semi-Supervised Learning. ICLR’23
>
> [3] Class-Distribution-Aware Pseudo-Labeling for Semi-Supervised Multi-Label Learning. NeurIPS’23
>
> [4] A Survey on Image Data Augmentation for Deep Learning. Journal of Big Data
>
> [5] Partial-Label Regression. AAAI’23

---

> ### Comment · Reviewer_RRwY · 2024-08-09
> **I will consider improving my score**
>
> I agree with what author mentioned that the significance and novelty of research should not be limited to algorithm design alone, proper summarization work is also equally important, even though this may not be as popular, especially in conferences.
>
> However, considering the author's coherent expression and the lack of similar work in this field, I would consider raising my score.
>
> I hope author can consider joining the discussion on PL regression. It seems that there has been some relevant work recently, such as Weakly Supervised Regression with Interval Targets. ICML'23.

---

> > ### Author Response · Authors · 2024-08-10
> > **Thank you for your positive comments**
> >
> > Thanks sincerely for your positive evaluation of our work, and we are very grateful that you are willing to raise your score.
> >
> > We'd like to express our sincere gratitude for your thoughtful and thorough comments. Thank you for the suggestion regarding an important task within regression contexts. We hope to contribute to this area of research in the future.
> >
> > Regarding the reception of summarization work at conferences, we understand that perspectives may vary. We note that top conferences like NeurIPS do welcome such submissions (a simple search for terms like "rethinking" or "understanding" within the titles of papers accepted at last year's NeurIPS yields more than 50 entries).

---

> > > ### Comment · Reviewer_RRwY · 2024-08-14
> > > **I would be happy to raise my score.**
> > >
> > > Thank you to the authors for their response. I would be happy to raise my score to 6.

---

> > > > ### Author Response · Authors · 2024-08-14
> > > >
> > > > Thanks a lot for your constructive suggestions!

---

### Official Review · Reviewer_TEwD · 2024-07-11

**Soundness:** 3
**Presentation:** 3
**Contribution:** 2
**Rating:** 6
**Confidence:** 4

**Summary:**

The paper presents a comprehensive empirical analysis of various Partial-Label Learning (PLL) methods. The authors identify that mini-batch Partial-Label (PL) purification is a key component for achieving top performance in PLL, as it progressively transitions from uniform to one-hot pseudo-labels. The study also introduces a minimal working algorithm that is simple yet effective, emphasizing the importance of mini-batch PL purification and suggesting improvements to enhance PLL methods​

**Strengths:**

1. The paper provides a detailed analysis and summary of the development in the PLL field and the core contributions of various state-of-the-art methods. I believe this is highly insightful and educational for future work.

2. The proposed StreamPurify is simple yet effective, achieving improvements over previous methods across different benchmarks.

3. The overall writing of the paper is very coherent and logical, with a clear analysis of the field's development and issues.

**Weaknesses:**

1. The paper attempts to extract commonalities from numerous methods to identify the most critical elements for PLL. The authors introduce a descriptive concept/definition called Mini-batch PL purification to summarize the label disambiguation process. However, this definition is limited to a textual description and does not effectively use mathematical language to depict it. For example, I think entropy or relative entropy (or their variants) could be a worthwhile choice to describe the label disambiguation process. When model prediction confidence is evenly distributed among multiple label candidates, the entropy of the label confidence distribution is high. However, when a winner label emerges, taking most of the prediction confidence, the entropy decreases. [1] has already analyzed and explored this from the perspective of entropy. In summary, failing to systematically and scientifically construct a mathematical definition of Mini-batch PL purification is the main drawback of this paper.

2. On the other hand, I believe many PLL works have realized that the key to label disambiguation is the process that is highly related to continuously increasing the max model prediction confidence. However, the main question is, during this process, how can we ensure that the model's high-confidence label is indeed the true label?

3. The authors provide another insight, which is that a label with high confidence tends to receive a larger gradient, further enhancing its advantage, i.e., the winner-takes-all phenomenon [2]. So, during model initialization, if uniform labels are used for warm-up training, wouldn't the label confidence distribution of a data instance be heavily influenced by the initial parameters? If some labels happen to have higher confidence at initialization, wouldn't their advantage continue to grow?

[1] Candidate-aware Selective Disambiguation Based On Normalized Entropy for Instance-dependent Partial-label Learning (ICCV 2023)

[2] Towards Unbiased Exploration in Partial Label Learning

**Questions:**

Please refer to the weaknesses section.

**Limitations:**

Yes, the authors provided the discussion of the limitations.

---

> ### Author Rebuttal · Authors · 2024-08-07
>
> We would like to thank the reviewer for taking the time to review our work and the positive comments regarding its significance and writing quality. Below are our responses to the comments in Weaknesses and Questions.
>
> > C1. The authors introduce a descriptive concept/definition called Mini-batch PL purification to summarize the label disambiguation process … failing to systematically and scientifically construct a mathematical definition of Mini-batch PL purification is the main drawback of this paper.
>
> Thank you for your insightful comments! Since our contributions center on understanding existing SOTA PLL methods, which themselves always lack a solid mathematical foundation, extracting commonalities from these methods requires a high degree of abstraction **from a philosophical and conceptual perspective**, even necessitating the omission of detailed methodological and algorithmic specifics. Therefore, establishing a solid mathematical foundation becomes nearly impossible. Additionally, it's important to note that the performance and runtime of mini-batch PL purification depend on certain hyperparameters, such as batch size. This dependency further reduces the applicability of a mathematical characterization.
>
> Thank you for your suggestion regarding entropy. As an alternative method to the widely used loss value [1], entropy is a heuristic technique for estimating the quality of pseudo-labels. To our knowledge, neither entropy nor loss allows for a strict mathematical definition of their mapping to label quality.
>
> > C2. On the other hand, I believe … high-confidence label is indeed the true label?  & The authors provide another insight … wouldn't their advantage continue to grow?
>
> Thank you for your insights! These are very good questions which are **critical issues in the PLL research area**, and all previous PLL works ought to have answered but often have not. First, we must clarify that although these issues are indeed foundational, they **serve as prerequisites rather than the main focus** of our paper. However, we appreciate the opportunity to humbly offer analysis on these concerns below:
>
> > C2Q1. I believe many PLL works have realized that the key to label disambiguation is the process that is highly related to continuously increasing the max model prediction confidence … how can we ensure that the model's high-confidence label is indeed the true label?
>
> We agree that the key to label disambiguation is continuously increasing the max confidence. This very process underscores the fact: we *cannot guarantee* that high-confidence label is indeed the true label, hence a progressive transition is essential.
>
> Then we would like to discuss why using the high-confidence label as the true label often works. This idea was first introduced by PRODEN [3] to the PLL community, and it has been extensively used in the noisy-label learning field (e.g. [1,4]). Its applicability in PLL due to the key assumption that the probability of a true label to be the candidate label is dominant. In fact, in the experimental setup of all PLL methods, it is assumed that $p(y \in S|x,S)=1, p(i \in S|x,S)<1, \forall i\neq y$. This implies that within any geometric neighborhood in the data space, the true label is dominant. Informally speaking, when stochastic gradient optimization is used, the true label contributes more frequent to the objective, making it more likely to be learned first. Then the PL purification process reinforces this knowledge. For more analysis related to this topic, we recommend reading [5].
>
> > C2Q2. The authors provide another insight, which is that a label with high confidence tends to receive a larger gradient, further enhancing its advantage, i.e., the winner-takes-all phenomenon [2].
>
> We must clarify that we *have not claimed* that “a label with high confidence tends to receive a larger gradient”. This statement originates from [2]. With all due respect, we disagree with this statement. In fact, the opposite is typically true, a property determined by the link function and the proper loss [6]. Higher confidence corresponds to lower loss and consequently smaller gradients.
>
> > C2Q3. If uniform labels are used for warm-up training, wouldn't the label confidence distribution of a data instance be heavily influenced by the initial parameters?
>
> Thank you for asking! Actually, label confidence distribution *will not* be heavily influenced by the initial parameters. Mathematically speaking, given any neural network, the parameters $w$ are typically initialized using a standard random normal distribution. According to the Chebyshev inequality and the Central Limit Theorem, we have: $P(|\bar{w}|≥a)≤\frac{\sigma^2}{na^2}$. When the number of neurons $n$ is sufficiently large, the sampled values of $w$ are very close to zero. Consequently, the output for each dimension $z=\max(0,wx+b)$ will be close to a small constant $b$. This means that during the initialization phase, no labels will receive significantly higher confidence. We conduct a toy experiment to demonstrate this point. We initialize different sizes of MLPs using various initialization methods and record their outputs for FMNIST after softmax (without training), specifically the margin between the highest and lowest outputs:
>
> | network\initialization | normal | Xavier | He |
> | --- | --- | --- | --- |
> | mlp-3 | 2.4e-3 | 0.03 | 0.05 |
> | mlp-5 | 4.9e-5 | 0.004 | 7.9e-3 |
> | mlp-10 | 2.4e-5 | 2.4e-5 | 2.0e-4 |
>
> We can observe the initial output distribution closely approximates a uniform distribution. Moreover, by using uniform targets to update for at least one epoch, we ensure that the model acquires a certain level of discriminative capability before relying on its outputs. Thank you for raising this point, we will add the discussion in the revised version.

---

> > ### Comment · Reviewer_TEwD · 2024-08-12
> >
> > Thanks for your response and corresponding analysis! The authors have solved most of my concerns. I will raise my score.

---

> > > ### Author Response · Authors · 2024-08-12
> > > **Thank you**
> > >
> > > Thank you very much for raising the score!
> > >
> > > We'd like to express our sincere gratitude for your thoughtful and thorough review of our paper.

---

> ### Author Response · Authors · 2024-08-07
> **Response to Reviewer TEwD (2/2)**
>
> > C2Q4. If some labels happen to have higher confidence at initialization, wouldn't their advantage continue to grow?
>
> This is not the case. The conclusions in this paper directly address this issue: mini-batch PL purification ensures the model's robustness to pseudo-labels of model’s outputs. We additionally add an experiment on FMNIST with instance-dependent PLs, performing a warm-up with uniform targets for 2/50 epochs, followed by training with our proposed minimal working algorithm SASM for the remaining 98/50 epochs. We record the improvement in accuracy of initially misclassified samples (the epoch in which the highest improvement achieved). As a comparison, we use the maximum confidence label predicted after the warm-up phase as one-hot for subsequent training.
>
> | epoch divided\training strategy | SASM | one-hot |
> | --- | --- | --- |
> | 2---98 | 69.16%(100ep) | 4.48%(67ep) |
> | 50---50 | 61.89%(100ep) | 5.00%(52ep) |
>
> The results indicate that even after a long warm-up phase, where the model has begun to overfit all candidate labels, the mini-batch PL purification is still capable of correcting errors.
>
> [1] DivideMix: Learning with Noisy Labels as Semi-Supervised Learning. ICLR’20
>
> [2] Towards Unbiased Exploration in Partial Label Learning
>
> [3] Progressive Identification of True Label in Partial-Label Learning. ICML’20
>
> [4] Searching to Exploit Memorization Effect in Learning with Noisy Labels. ICML’20
>
> [5] Does Learning Require Memorization: A Short Tale about a Long Tail. STOC’20
>
> [6] Composite Binary Losses. JMLR

---

### Official Review · Reviewer_ygK7 · 2024-07-12

**Soundness:** 4
**Presentation:** 4
**Contribution:** 3
**Rating:** 7
**Confidence:** 5

**Summary:**

The paper offers a comprehensive empirical analysis to understand what makes partial-label learning (PLL) methods effective, focusing on the transition from uniform to one-hot pseudo-labels in mini-batch PL purification. It analyzes the complexity and diversity of SOTA PLL methods, proposing simplified algorithm design principles that maintain high performance with less complexity. It introduces a minimal working algorithm that adheres to the principles and shows its effectiveness.

**Strengths:**

The authors revisit PLL and provide a detailed empirical analysis that addresses its fundamental question—what contributes to the effectiveness of PLL methods. It might not only clarify the field but also guide future research directions. Writing is good quality.

**Weaknesses:**

The definition and implementation details of mini-batch PL purification are not sufficiently clear, particularly the impact of batch size on the effectiveness. It is unclear why this technique is the most effective points.

The limitations of the proposed principles do not be discussed, particularly in terms of the practical applicability of the introduced algorithms like SASM, SADM, DASM, and DADM. Future guidelines on how to effectively utilize these methods would be beneficial.

Theoretical results would provide a stronger validation of the proposed principles.

Comparisons on real-world datasets would be helpful.

**Questions:**

See above

---

> ### Author Rebuttal · Authors · 2024-08-07
>
> We would like to thank the reviewer for taking the time to review our work and the positive comments regarding its significance and writing quality. Below are our responses to the comments in Weaknesses and Questions.
>
> > C1. The definition and implementation details of mini-batch PL purification are not sufficiently clear, particularly the impact of batch size on the effectiveness. It is unclear why this technique is the most effective points.
>
> Thank you for raising this point. We give a detailed analysis of the advantages mini-batch PL purification and also conduct a sensitivity analysis on the batch size. Please kindly refer to our response to Comment 2 of Reviewer PELT.
>
> > C2. The limitations of the proposed principles do not be discussed, particularly in terms of the practical applicability of the introduced algorithms like SASM, SADM, DASM, and DADM. Future guidelines on how to effectively utilize these methods would be beneficial.
>
> Thank you for the comment. Firstly, we need to clarify that all the proposed methods in this paper serve more as demos to illustrate our findings.
>
> **Practical applicability** We compare these methods on different PL datasets. Specifically, we set the instance-independent flipping probability $\eta={0.3,0.7,0.9}$, class-dependent flipping probability $\eta={0.6}$, and also generate the instance-dependent PLs on FMNIST. The results of test accuracy are as follows:
>
> | method\PLgeneration | ins.-indep0.3| ins.-indep0.7| ins.-indep0.9| class-dep0.6 | ins.-dep. |
> | --- | --- | --- | --- | --- |--- |
> | SASM	| 93.83 | 92.18 | 90.12 | 94.12 | 90.64 |
> | SADM	| **94.02** | 92.36 | 90.10 | 94.10 | 90.65 |
> | DASM	| 93.89 | **92.85** | **90.97** | 94.15 | 91.00 |
> | DADM | 93.59 | 92.40 | 90.54 | **94.23** | **91.04** |
> | PRODEN+| 93.70 | 92.52 | 88.27 | 94.09 | 89.75 |
>
> We can observe that different methods exhibit variations in performance under different scenarios, particularly the versions with multiple augmentations. It is important to note that we did not delve into many tuning details because outperforming SOTA methods is not our objective. Further, the simplified methods can also serve as backbones for incorporating stronger components and provide important prototype for future exploration into PLL. An intuitive idea is that stronger components are expected to achieve better performance, and we believe it is also necessary to consider the coupling effects between different components. The instantiation details are not within the scope of this paper and we leave the explorations to future work.
>
> **Limitations** As mentioned in our paper, using simulated PLL versions of vision datasets is currently the standard practice for evaluating PLL methods. Consequently, our work follows this established approach. The explorations on various types of real-world datasets may be a critical direction for future research in the PLL field.
>
> > C3. Theoretical results would provide a stronger validation of the proposed principles.
>
> Thank you for your insightful comments. Since our goal is understanding effectiveness of SOTA PLL methods, which themselves always lack a solid mathematical foundation, extracting commonalities from these methods requires a high degree of abstraction **from a philosophical and conceptual perspective**, even necessitating the omission of detailed methodological and algorithmic specifics. Therefore, establishing a solid mathematical foundation becomes nearly impossible with the current mathematical tools. Additionally, it's important to note that the performance and runtime of mini-batch PL purification depend on certain hyperparameters, such as batch size. This dependency further reduces the applicability of a mathematical characterization.
>
> > C4. Comparisons on real-world datasets would be helpful.
>
> Thanks for the valuable suggestion. Since these datasets are in tabular form, comparing methods should be compatible with scenarios that do not involve data augmentation. All the methods use MLP-3 and batch size is 256. IPAL [1] is a most advanced method without mini-batch PL purification. The results of test accuracy are as follow. The results on real-world datasets support our conclusion.
>
> | method\dataset | Lost | BirdSong | MSRCv2 | Soccer Player | Yahoo! News |
> | --- | --- | --- | --- | --- |--- |
> | SASM | 69.84 | 69.38 | 53.84 | **54.55** | 64.99 |
> | SADM | 69.38 | **70.32** | **57.29** | 54.25 | 65.04 |
> | PRODEN | **73.60** | 70.12 | 56.60 | 54.33 | **67.23**|
> | IPAL | 65.75 | 70.30 | 53.47 | 53.36 | 65.55 |
>
> [1] Solving the Partial Label Learning Problem: An Instance-Based Approach. IJCAI’15

---

> > ### Comment · Reviewer_ygK7 · 2024-08-11
> > **Keep the scores**
> >
> > Thank you for your response.

---

> > > ### Author Response · Authors · 2024-08-14
> > >
> > > Thank you very much for your positive comments!

---

### Official Review · Reviewer_PELT · 2024-07-12

**Soundness:** 3
**Presentation:** 3
**Contribution:** 3
**Rating:** 7
**Confidence:** 4

**Summary:**

This paper simplifies the process of PLL by distilling critical success factors for high performance into a minimal algorithmic design through extensive empirical analysis. This work is a step forward in making PLL methods more accessible and efficient, offering substantial insights into traditional complex algorithms.

**Strengths:**

The paper is highly motivated, offering impressive reflections on the categorization and algorithm design principles in the PLL community. The paper's main contribution lies in enhancing understanding, and a simple improvement StreamPurify driven by this understanding, is commendable.

**Weaknesses:**

1. There are several simplified approaches that achieve comparable performance, which makes the conclusions less clear.

2. While the design principles ensure comparable performance, it remains unclear what implications this has for future algorithms, for example, how can stronger components be integrated with these design principles?

3. The effectiveness of the proposed mini-batch PL purification is supported only by some empirical results, and the significance of this change should be discussed in more detail.

**Questions:**

1. The content of the paper is overly dense and requires further clarification, making it somewhat challenging to read and understand.

2. In the proposed mini-batch PL purification, how sensitive is the performance to different batch sizes, and are there optimal batch size recommendations based on your experiments?

---

> ### Author Rebuttal · Authors · 2024-08-07
>
> We would like to thank the reviewer for taking the time to review our work and the positive comments regarding its significance. Below are our responses to the comments in Weaknesses and Questions.
>
> > C1. Q1. There are several simplified approaches achieve comparable performance, which makes the conclusions less clear. Q2. It remains unclear what implications this has for future algorithms, for example, how can stronger components be integrated with these design principles?
>
> Our main contributions are to abstract the consistent philosophy underlying SOTA PLL methods into minimal algorithm design principles, thus avoiding future algorithm designs based on trial and error. **Takeaway**: Designing well-performed PLL methods should purify PLs in a mini-batch-wise manner.
>
> The simplified methods mainly **serve as demos to illustrate our findings**. Thank you for your insight that they can further serve as backbones for incorporating stronger components and provide important prototype for future exploration into PLL. The intuitive idea is that stronger components are expected to achieve better performance, and we believe it is also necessary to consider the coupling effects between different components. Consider SASM and SADM and combine them with consistency regularization. We conduct experiments on FMNIST with instance-independent and instance-dependent PLs. The results of test accuracy are as follows:
>
> | method\flippingprobability | 0.3 | 0.7 | ins.-dep. |
> | --- | --- | --- | --- |
> | SASM | 93.83 | 92.18 | 90.64 |
> | + consistency regularization | 93.89 | **92.85** |  91.00 |
> | SADM | **94.02** | 92.36 | 90.65 |
> | + consistency regularization | 93.59 | 92.40 | **91.04** |
>
> We observe that directly combining a generally useful method does not necessarily boost performance. The results are related to the generation process and instantiations details (e.g. what augmentation involved). The specific algorithm design is beyond the scope of this paper. We leave the exploration of different instantiations of design principles and their combinations with various components to future work.
>
> > C2. The effectiveness of the proposed mini-batch PL purification is supported only by some empirical results, and the significance of this change should be discussed in more detail. Q2. In the proposed mini-batch PL purification, how sensitive is the performance to different batch sizes, and are there optimal batch size recommendations based on your experiments?
>
> Q1. Thank you for the constructive comments. Mini-batch PL purification was first proposed in [1], and widely adopted in subsequent PLL methods. The key difference from previous PLL methods lies in the frequency of estimating pseudo-labels and updating the model iterations. It is worth mentioning that mini-batch-wise update has been used in many other fields such as noisy-label learning [2], distribution shift [3], etc. We will give a detailed analysis of its advantages, and the relevant discussions have been in [1,2]. Traditional EM-based PLL methods update the model based on the current pseudo-labels until convergence. To avoid overfitting to pseudo-labels, epoch-wise PL purification uses pseudo-labels for model updates in the whole next epoch. The confirmation bias caused by inaccurate pseudo-labels accumulates throughout the epoch. In contrast, batch-wise PL purification updates the model right after estimating pseudo-labels of each batch. The updated model can better estimate pseudo-labels of the next batch, alleviating the confirmation bias batch by batch.
>
> Q2. We conduct a sensitivity analysis on the batch size on FMNIST with instance-dependent PLs. The test accuracy are as follows:
>
> | method\batchsize | 32 | 64 | 128 | 256 | 512 | #dataset |
> | --- | --- | --- | --- | --- | --- | --- |
> |SASM | 89.85 | 89.85 | 90.29 | **90.64** | 89.75 | 80.87 |
> |SADM | 89.86 | 89.91 | 90.14 | **90.65** | 89.56 | 78.02 |
> |DASM | 89.93 | 90.51 | 90.57 | **91.00** | 90.35 | 80.80 |
> |DADM | 90.16 | 90.01 | 90.46 | **91.04** | 89.99 | 78.53 |
> |PRODEN+ | 89.09 | 89.24 | **90.01** | 89.91 | 89.76 | 79.54 |
>
> We can observe that very coarse execution granularity (large batch size) might result in the model trapped in local optimum, while very fine granularity (small batch size) might result in unstable estimation of the true labels, leading to degraded performance. We recommend treating batch size as a hyperparameter that requires tuning for optimal performance. The batch sizes used in our experiments are shown in Appendix A, and will add the analysis in the revised manuscript.
>
> [1] Progressive Identification of True Labels for Partial-Label Learning. ICML’20
>
> [2] SemiNLL: A Framework of Noisy-Label Learning by Semi-Supervised Learning. TMLR
>
> [3] Rethinking Importance Weighting for Deep Learning under Distribution Shift. NeurIPS’21

---

> > ### Comment · Reviewer_PELT · 2024-08-11
> > **The authors have addressed all of my concerns.**
> >
> > Reviewing all the comments and responses, I thank the authors for their detailed rebuttal. I find the authors' responses to be sufficient, and they have properly addressed all my concerns.
> >
> > The paper is good, its significance is solid, and I have raised my score to 7.

---

> > > ### Author Response · Authors · 2024-08-14
> > >
> > > Thank you very much for dedicating your time to reviewing our paper and raisng the score!

---

### Decision · Program_Chairs · 2024-09-25

**Decision:**

Accept (poster)

**Comment:**

This is an interesting and solid contribution on partial-label learning (PLL), which investigates what makes a PLL method effective. All reviewers were positive about the paper and recommended accepting the submission.